# The Importance of Magnetic Resonance Enterography in Monitoring Inflammatory Bowel Disease: A Review of Clinical Significance and Current Challenges

**DOI:** 10.3390/diagnostics15121540

**Published:** 2025-06-17

**Authors:** Roxana Elena Mirică, Teodora Florentina Matură, Eliza Craciun, Dana Pavel

**Affiliations:** 1Department of Social Insurance Medicine, Faculty of Medicine, “Carol Davila” University of Medicine and Pharmacy, 020021 Bucharest, Romania; 2National Institute of Medical Expertise and Recovery of Work Capacity, 050659 Bucharest, Romania; 3Department of Gastroenterology, Private Healthcare Network, Regina Maria, 011603 Bucharest, Romania; 4Department of Radiology—Medical Imaging, Private Healthcare Network, Regina Maria, 077190 Bucharest, Romania; teodora.matura@reginamaria.ro; 5Department of Radiology—Medical Imaging, Private Healthcare Network, Regina Maria, 011603 Bucharest, Romania; eliza.craciun@reginamaria.ro; 6Department of Radiology—Medical Imaging, Private Healthcare Network, Euroclinic Hospital—Regina Maria, 014461 Bucharest, Romania; dana.pavel@reginamaria.ro

**Keywords:** MR enterography, inflammatory bowel disease, colonoscopy, intestinal ultrasound, imaging

## Abstract

Inflammatory bowel diseases are chronic diseases of the gastrointestinal tract with a growing prevalence worldwide, requiring precise diagnostic and monitoring methods to guide their appropriate treatment. In this context, MRE (Magnetic Resonance Enterography) has become an essential imaging technique as a non-invasive option for the diagnosis of Crohn’s disease and ulcerative colitis in recent years. This method provides detailed information about intestinal inflammation, disease activity, complications, and response to therapy, without the need to expose the patient to ionizing radiation. This study analyzes the advantages of MRE over other imaging methods, as well as its clinical applicability and current challenges. We also discuss future perspectives, including the integration of artificial intelligence and the optimization of protocols for better diagnostic accuracy.

## 1. Introduction

Inflammatory bowel diseases are a heterogeneous group of chronic, disabling gastrointestinal disorders characterized by inflammation of the digestive tract, including ulcerative colitis and Crohn’s disease [1,2]. Ulcerative colitis affects the entire colon, with inflammation limited to the mucosal layer, while Crohn’s disease can affect any segment of the digestive tract with transmural involvement [3].

Inflammatory bowel diseases are a global problem with a significant impact on the quality of life of affected patients, requiring effective diagnostic and monitoring strategies. In developed countries, while the incidence of these two diseases has stabilized, their prevalence is increasing: approximately 1.6 million people are affected in the USA while, in Europe, almost 3 million are affected [3,4]. The age of onset of both ulcerative colitis and Crohn’s disease is between 15 and 30 years, although there is a second peak between 50 and 80 years [5,6]. The etiologies of inflammatory bowel diseases have not been fully elucidated, but several triggering factors are involved, including genetic and environmental triggers, as well as alterations in the gut microbiota and intestinal immune system as a result of an inadequate response to immune hyper-reactivity [3,7].

According to the ECCO-ESGAR (European Crohn’s and Colitis Organisation–European Society of Gastrointestinal and Abdominal Radiology) guidelines, the diagnosis of inflammatory bowel diseases is based on the correlation of clinical features, laboratory tests, endoscopic, histological (endoscopic and histological findings), and imaging data [3,8].

Although colonoscopy with biopsy remains the gold standard for identifying mucosal lesions, this procedure is invasive and cannot provide information about intestinal segments located outside its visualization range, especially the small intestine [9]. In this context, imaging techniques, including intestinal ultrasound (IUS), MRE (Magnetic Resonance Enterography), and CT (computed tomography), play essential roles in assessing the extent of the disease, detecting complications, and guiding treatment [10]. Among the available imaging methods, MRE has established itself as a first-line technique for exploring the small intestine and colon, providing detailed functional and structural information, highlighting the activity and severity of inflammatory bowel diseases, staging transmural lesions [11], and evaluating therapeutic responses [12]. At the same time, MRE is indicated in patients with normal colonoscopy results [13].

This review aims to analyze the utility of MRE in the diagnosis and monitoring of inflammatory bowel diseases, highlighting its advantages, current challenges, and future perspectives.

## 2. Imaging Methods Used in Inflammatory Bowel Diseases

The diagnosis and monitoring of inflammatory bowel diseases requires precise imaging methods which are capable of evaluating both mucosal lesions and inflammatory changes in the intestinal walls and perenteric tissues.

Colonoscopyremains the gold standard for diagnosis, allowing for direct visualization of the mucosa, biopsy sampling [9], and treatment of some complications. It is superior to other imaging methods in highlighting superficial erosions and ulcerations, mucosal hyperemia, and loss of vascular pattern and detecting colonic polyps [2]. Despite its advantages, it has certain limitations: it is invasive, can be uncomfortable for the patient, and does not allow for visualization of the entire small intestine, requiring the use of additional imaging techniques such as CT, intestinal ultrasound, and MRE [14]. However, a colonoscopic evaluation is necessary if patients have persistent symptoms despite normal MRE results [15]. Colonoscopy also cannot evaluate extraintestinal lesions and may limit the penetration of the endoscope in the presence of stenosis (stricture) [16]. Additionally, during the examination, lesions located in hidden parts of the colon may be missed [17].Video capsule endoscopy is useful in exploring the small intestine, with high sensitivity for early mucosal changes, such as small aphthous lesions [18]. It is also indicated in patients with suspected Crohn’s disease and normal endoscopic results [19,20]. Recent studies have shown that video capsule endoscopy is superior to MRE in detecting lesions located in the proximal region of the small intestine [19]. However, it does not allow for in-depth evaluation of the intestinal wall and carries a risk of capsule retention in stenoses and intestinal obstructions.Intestinal ultrasound is a non-invasive, accessible, radiation-free method that does not require prior preparation, except for fasting a few hours before the examination [8]. Firstly, it allows for the detection of wall thickening, with a value over 3 mm considered pathological [8] and a measurement over 7 mm indicating an unfavorable prognosis, with surgical indication within the following year [21]. At the same time, the use of color Doppler or contrast medium (CEUS) allows for evaluation of both the wall perfusion and the intestinal inflammatory status, as well as the presence of complications (fistulas, abscesses, or inflammatory lesions), visualized as hypo- or hyperechogenic masses [8]. An increased color Doppler signal is observed in cases of transmural edema present in the active form of Crohn’s disease, as evidenced by disrupted mural stratification [22]. Intestinal ultrasound is also useful in detecting the thickening of peri-visceral adipose tissue or fat wrapping [23], as evidenced by increased echogenicity at this level, representing a sign of active disease [22]. Ultrasound images may be unsatisfactory and limited in obese patients, body habitus, or significant abdominal distension that may obscure the intestinal region [22].Computed tomography enterography provides detailed images of both the small intestine—highlighting intestinal wall thickening, hyperemia, submucosal fat deposition, and lymphadenopathy [24,25]—and of extraintestinal, perineural lesions, with greater accuracy in terms of the degree and severity of the disease [26], differentiating the active form from the fibrotic one. At the same time, this imaging technique is frequently used for the detection of complications of inflammatory bowel diseases (fistulas, perforations, and abscesses) [24,25] and in emergencies, such as sepsis or penetrating intra-abdominal lesions requiring surgical intervention [13]. Other advantages of CTE include a shorter scanning time, reduced costs compared to MRE [27], and suitability for patients with contraindications to MRE [13], those who are allergic to gadolinium-based contrast media [8], those who were claustrophobic in prior MR exams, and those with acute symptoms [13]. The main disadvantage is exposure to ionizing radiation, which limits its repeated use in young patients [28,29]. The radiation dose used in CTE for the adult population is between 10 and 20 mSv (milisievert) [30], while that in the pediatric population is between 2.9 and 4 mSv [31]. New protocols propose reducing the radiation dose in adults to 5–7 mSv and the noise produced by CTE during the investigation [32]. At the same time, recent studies have focused their interest on artificial intelligence and radiomics. Li et al. have demonstrated that a radiomics model (RM) based on CTE accurately describes intestinal fibrosis in patients with CD [33].Magnetic Resonance Enterography (MRE) is essential in the evaluation of inflammatory bowel disease, providing simultaneously detailed images of the intestinal wall and adjacent structures and inflammatory lesions [22], differentiating inflammation from fibrosis in both the small and large intestine submucosa and the perineal area [34,35]. MRE also has high accuracy in staging small bowel inflammatory bowel disease [29], in monitoring treatment response and relapse [22], and in detecting and classifying isolated forms of colonic involvement [36]. This imaging modality is preferred in complex cases with evidence of penetrating, fistulizing, and stenosing lesions [22], as well as in fistulas and perianal sepsis [13]. Fat smudging, fecal sign, fluid level, gaseous distension, comb sign (related vascular congestion), and lymphadenopathy are the elements mainly visualized/detected by MRE [2]. Another advantage—perhaps the most important—is that MRE is the safest and most cost-effective cross-sectional imaging method that can be used to evaluate the activity of Crohn’s disease and ulcerative colitis in both adults and young people [37], without the use of ionizing radiation [2]. Taylor et al. have shown that MRE has a sensitivity of 97% for detecting inflammatory bowel diseases, over 90% for fibro-inflammatory strictures, and specificity of over 95% [29].

## 3. Technical Principles of MRE

For optimal evaluation, MRE requires a standardized protocol, including the administration of an oral contrast agent and the use of T2 and diffusion-weighted imaging (DWI) sequences to highlight active inflammation.

### 3.1. Standardized Protocol for MRE

A strict protocol is followed to obtain quality images, which includes the following steps:

(a) Patient preparation before the examination involves following a light diet 24 h before, fasting 4–6 h before, and light hydration with water up to 1–2 h before the investigation. Fractional administration (250 mL/10–15 min) of 1–1.5 L within 45–60 min of an oral contrast agent—for example, hypo-osmolar or iso-osmolar solutions such as Mannitol 2.5%, PEG (polyethylene glycol), or water with methylcellulose—is necessary for adequate distension of the intestine prior to the scan. Complete intestinal emptying with purgative solutions is not necessary, as MRE aims at distending the loops with liquid, not complete emptying, and these solutions may produce artifacts through hyperperistalsis or intestinal irritation [2,38]. Additionally, 10–15 min before the acquisition, glucagon (0.5–1 mg im or iv) or butylscopolamine (20 mg iv) is administered to reduce bowel movements [39].

(b) Technical parameters—imaging protocol: The recommended equipment is a 1.5 Tesla or 3 Tesla MRI. The patient is asked to lie on the bed in the supine position.

T1 and T2 sequences with fat suppression, as dynamic sequences after the administration of iv contrast (Gadolinium) and diffuse light (DWI) sequences, are obtained. The images acquired in the axial and coronal planes allow for a detailed visualization of the intestinal wall and perineal tissues. The total duration of the examination is 30–45 min.

T2-weighted HASTE (half-Fourier single-shot turbo spin-echo)/SSFSE (single-shot fast spin-echo sequences): Axial and coronal (with or without fat suppression), with a TR (repetition time)/TE (echo time) of 90 ms, flip angle 90°, matrix 320 × 320, and thickness of 4–6 mm. This is recognized as one of the most important sequences for the evaluation of the small intestine and aims to highlight detailed intestinal and mesenteric structures, extraintestinal regions, and parietal inflammation and edema [40,41].

Balanced coronal steady-state free procession gradient-echo (SSFPGR)is used to evaluate the intestinal walls, mesenteric structures, and ganglia [41].

Three-dimensional cinematic coronal bSSFP captures extremely fast images of intestinal peristalsis in realtime [42].

Dynamic coronal 3D T1-weightedhigh-resolution isotropic volume (THRIVE)—GRE (gradient recalled echo) with FS (fat suppression) post-contrast allows for the evaluation of contrast uptake (inflammation versus fibrosis); matrix 256 × 256, thickness 2 mm, flip angle 10°. Sequences are obtained in multiple series post-injection of antiperistaltic agents, within 40–55 s for the enteric phase and 60–70 s in the portal venous phase, as well as at 180 s [2,8].

Delayed axial 3D T1-weighted post-contrast fat-saturated GRE (gradient recalled echo)allow for evaluation of complications such as fistulas and abscesses [43].

The DWI (diffusion-weighted imaging) sequence is obtained with the aim of detecting active inflammation, fibrosis, edema, and vasculopathy. It includes b-values of 0.400 and 800 s/mm^2^, which reflect the sensitivity of the image to Brownian motion; the higher the value, the more sensitive the image to diffusion. It also includes ADC, with an acquisition time of approximately 4 min. Inflamed areas present DWI hypersignal with low ADC (apparent diffusion coefficient) in active areas. The ADC value is insufficient for assessing the response to treatment if used alone [44] (Table 1).

### 3.2. Relevant Imaging Features in Magnetic Resonance Enterography

The most important imaging findings indicative of disease activity are mucosal hyperenhancement/hyperemia, wall thickening, intramural edema, ulcerations, lymphadenopathy, vascular congestion of mesenteric arteries, and strictures [2,8] (Table 2).


Mural thickening:Can be mild (<5 mm), moderate (<9 mm),orsevere (>10 mm).Commonly occurs in active areas of inflammation (Figure 1).Mural hyperenhancementAsymmetric distribution in CD or continuous and concentric distribution in extensive ulcerative colitis [43].Stratified uptake: “double layer” (submucosa is thickened by edema and inflammation) or “trilaminar layer” (when serosa is also involved) [8].Homogeneous, hypovascular uptake in (chronic) fibrosis.Correlates with clinical and biological activity scores [43].T2 hypersignal and rapid contrast enhancement compared to adjacent intestinal areas [46,47].Evaluated on post-gadolinium T1 fat-sat sequences, in dynamics [48].Intramural edemaIs detected as T2 hyperintense signals.Restricted diffusionDWI hypersignal + low ADC in acute inflammation.Allows for differentiation of inflammation from fibrosis [48] (Figure 2 and Figure 3).UlcerationFocal defects, fine disruptions of the mucosal contour—small signs of T2hyposignal or intense post-contrast enhancement [48].Requires adequate distension of the small bowel (Figure 4).Mesenteric lymphadenopathyEnlarged mesenteric lymph nodes, over 5 mm transverse diameter [49] and >1 cm axial diameter [43].Non-specific, but supports active inflammation [42] (Figure 5).Comb signDilatated and elongated vasa recta (vascular congestion of the mesenteric arteries) [44] (Figure 6).Fibrofatty proliferationAlso called “creeping fat”.Sometimes inflamed with a “mesenteritis” appearance; formation adjacent to the inflamed intestinal region with a characteristic signal of fibrous or fatty consistency [44] (Figure 5).FistulasThey may be enteroenteric, enterocolic, enterovesical, or perianal [8].Fistulae occur following advanced penetrating disease [8].AbscessesAbscesses are found in the abdominal cavity, intestinal wall, or perianal area [8].StenosisReduced caliber by more than 50% compared to normal intestinal loops [8], with upstream loop dilatation ≥ 3 cm (mild) or > 4 cm (moderate–severe) [8].May be inflammatory (with edema and entrapment) or fibrotic (without inflammatory signs) (Figure 7).


**Table 2 diagnostics-15-01540-t002:** Relevant imaging features in Magnetic Resonance Enterography.

Characteristic	Appearance in MRE	Clinical Significance
Mural thickening	A value of >3 mm is considered pathological [50] T2 hypersignal	Active inflammation or fibrosis [43]
Mural hyperenhancement/hyperemia	T2 hypersignal and rapid contrast enhancement	Active inflammation [43] Disease activity assessment
Intramural edema	T2 hypersignal	Active inflammation [43]
Fibrosis	T2hyposignal and progressive contrast enhancement	
Restricted Diffusion (DWI/ACD)	DWI hypersignal + low ADC in acute inflammation	Active inflammation
Ulceration	Focal defects, T2hyposignal, or intense post-contrast enhancement [48]	Sign of severe activity
Mesenteric lymphadenopathy	T2 hyperintense signal and enhanced contrast	Regional inflammation [43]
Comb sign	Vascular congestion of the mesenteric arteries [44]	Active disease, intestinal inflammation present
Fibrofatty proliferation	Presents intermediate T1 signal on MRE and is of mesenteric origin [44]	Active inflammation
Fistulas	T2-hyperintense, post-contrast enhancing tracts, sometimes with adjacent collections [47]	Transmural complication
Abscess	Fluid collections, encapsulated, with an absorbing and restrictive periphery on DWI [47]	Infection, complication, requires treatment
Stenoses	Areas of thickening with prestenotic dilatation [47]	Complication Evaluation for intervention
Obstructions	Complete obstruction of the intestinal lumen [47]	

## 4. Applicability of MRE in Inflammatory Bowel Diseases

MRE has become established in the last decade and is widely used in clinical practice for the differential diagnosis between Crohn’s disease and ulcerative colitis [43]. As a non-irradiating, reproducible, and versatile method, this imaging technique provides detailed information about the intestinal wall and adjacent structures and allows for the early detection of complications such as fistulas and abscesses, with an accuracy comparable to that of exploratory surgery [28]. In addition, MRE plays a crucial role in monitoring the response to treatment, with studies demonstrating a direct correlation between imaging changes and the clinical evolution of patients treated with biological agents.

MRE allows for characterization of the affected segments and establishment of the transmural distribution of inflammation. It is sensitive in detecting lesions in the small intestine—a region which is difficult to access when using conventional colonoscopy. It may reveal thickening of the intestinal wall, submucosal edema, deep ulcerations, or segmental and saltatory extension (specific to Crohn’s disease) or continuous lesions (in cases of ulcerative colitis).

### 4.1. Differentiating Between Active Inflammation and Fibrosis

One of the most valuable contributions of MRE is the ability to differentiate between active inflammation (characterized by edema, layered contrast enhancement, and increased DWI signal) and fibrosis (with homogeneous enhancement, without edema or restrictive signal on DWI). This distinction is crucial in medical versus surgical therapeutic decisions. A recent study has suggested that fibrotic lesions seen on MRE can be confirmed histologically, with approximately 98% diagnostic accuracy [51]. Another study has shown that the performance of MRE decreased in situations where active inflammatory lesions and intestinal fibrosis coexisted [52].

### 4.2. Screening/Detection of Complications

MRE is excellent in identifying typical complications of inflammatory bowel diseases, especially Crohn’s disease:
➢Enteroenteric, enterocutaneous, and perianal fistulas: MRE can distinguish between simple and complicated fistulas, guiding the decision between conservative treatment and surgical drainage;➢Intra-abdominal abscesses;➢Fibrous stenosis with dilation of the upstream loops;➢Mesenteric adenopathies and changes in perenteric fat;➢Toxic megacolon—a rare but severe complication of ulcerative colitis [28].

### 4.3. Monitoring Response to Treatment

MRE is ideal for assessing the efficacy of biological treatment and immunosuppressive or immunomodulatory therapies. The reduction in wall thickness, the disappearance of edema, and a decrease in post-contrast uptake reflect the therapeutic response and can predict clinical remission. In clinical studies, the concept of “mucosal healing” represents an indicator associated with sustained clinical remission and a reduced rate of surgical interventions and hospitalizations [53,54,55,56], making it a therapeutic target in patients with inflammatory bowel diseases [54,57]. Recent studies have suggested that MRE can detect mucosal healing. Imaging changes can be used to adjust treatment before the onset of clinical symptoms, minimizing the adverse effects of expensive drugs by delaying their administration [58].

### 4.4. Complementarity with Other Methods

While colonoscopy remains the standard for colonic mucosal visualization and biopsy, MRE is indispensable for the following:
➢Exploration of jejunal and ileal loops;➢Transmural and extramural evaluation;➢Therapeutic guidance in the absence of obvious colonic lesions.

### 4.5. Role in Staging and Imaging Scores

Through the use of standardized scores (MaRIA, sMaRIA, London, and Nancy), Magnetic Resonance Enterography provides an objective and reproducible assessment of the severity of inflammation. This is useful in both clinical and research settings [13].

#### 4.5.1. Standardized Imaging Scores in Magnetic Resonance Enterography

MRE allows for the quantification of intestinal inflammation through validation scores, such as MaRIA, sMaRIA, London, Nancy, Clemont, and MEGS, which are useful for standardization, longitudinal follow-up, and assessing responses to biological therapies. They generally measure histologic and endoscopic inflammation, and/or quantitative measurements [13], including wall thickening, mural edema, ulceration [22], intramural hyperintense T2 signal, and relative contrast enhancement [13], in comparison to an index tissue (e.g., a normal bowel wall or psoas muscle) [3].

1.The most widely studied scoring system that assesses Crohn’s disease activity on MRE is the magnetic resonance index activity (MaRIA) score. The score is calculated using the following equation:MaRIA score=1.5 × wall thickness + 0.02 × RCE (relative contrast enhancement) + 5 × edema + 10 × ulceration [59];RCE = [(WSI − wall signal intensity postgadolinium − WSIpregadolinium)/(WSIpregadolinium)] × 100 × SD noise pregadolinium/SD noise postgadolinium), where SD (standard deviation) noise represents the average of three SDs of the signal intensity measured outside the body before and after administration of the contrast agent [59].The cut-off values of the MaRIA score are as follows:

➢Normal: 0–6;➢Moderate disease: ≥ 7–11;➢Severe disease: ≥ 11 [22].

2.The major disadvantage of this score is that it is time-consuming to obtain. Such a limitation led to the development of a simplified new scoring system, the sMaRIA, which requires just 4.5 min compared to over 12 min for the MARIA [22,59,60]. The sMaRIA was validated by Ordas et al. in 2019, and its most significant advantage is that it does not involve contrast-enhanced imaging [22,61].The sMaRIA is calculated with the following equation:MARIAs = (1 × thickness >3 mm) + (1 × edema) + (1 × fat stranding) + (2 × ulcers)The cut-off points of the sMaRIA score are as follows:

➢A score of >1 identifies active disease, with 90% sensitivity and 81% specificity;➢A score of >2 indicates severe lesions, with 85% sensitivity and 92% specificity [22,61].

A recent study has confirmed the accuracy of the sMaRIA score, which identified endoscopic remission after 3 months of treatment with anti-TNF α agents and corticosteroids [22].

3.In 2014, the Crohn’s disease activity score (CDAS) was modified into a global score called the MR enterography global score (MEGS) [43], which evaluates the entire small and large bowel [62].

The total MEGS is a sum of scores of qualitative and semiquantitative assessments: score per segment × multiplication score per segment + additional score per patient (lymph node, comb sign, abscess, and fistula) [43,63] (Table 3).

Makanyanga et al. compared the MEGS with laboratory biomarkers of activity, such as fecal calprotectin > 100 µg/g [64]. The calculation formula involved the elevated fecal calprotectin value and the probability of active disease in patients with Crohn’s disease. It was formulated using a logarithmic form: *α* = 1.8 × wall thickness + 0.08 × mural T2 + 0.19 × length − 0.192 [64].

#### 4.5.2. London and “Extended” London Scores

In 2012, Steward et al. validated the London and “extended” London scores using the endoscopic acute inflammatory score (AIS), a histopathological grading of the terminal ileum as the reference standard [43,65], with a sensibility of 81% and specificity of 70% [22,65].

The London score is calculated as follows: 1.79 + 1.34 × mural thickness score + 0.94 × mural T2 score [63].

The “extended” London score is a detailed version and assesses both active inflammation and complications of Crohn’s disease. The major disadvantage is that it requires a contrast agent (gadolinium), which represents a limitation in clinical practice. As such, it is mainly applied in research trials [22].

#### 4.5.3. Crohn’s Disease MRI Index (CDMI)

The CDMI was validated in 2012, with the objective of quantifying inflammatory activity on Magnetic Resonance Enterography. It is a semiquantitative, rigorous, and detailed score with good histological correlations. As it requires precise measurements and specific software for calculating the gradient, it is less commonly applied in clinical practice [63].

Although recent literature references have equated the CDMI with the “extended” London score, the two have different methodologies and bodies. While the CDMI involves post-contrast gradient measurements, the London score and “extended” London score are based on visual assessment of morphological and functional parameters.

#### 4.5.4. Clermont Score

The Clermont score is another MRE index which was validated in 2013, representing an efficient tool for the evaluation of CD activity. It is similar to the MaRIA score [66] in terms of the parameters highlighted (wall thickness, edema, and ulceration) [63]. The Clermont score uses ADC measurements from diffusion-weighted sequences [67].

The Clermont score is calculated as follows: −1.321 × ADC (mm^2^/s) + 1.646 × wall thickness (mm) + 8.306 × ulcers + 5.613 × edema + 5.039 [43,63], with a sensitivity of about 82% and specificity of 100% [67].

#### 4.5.5. Nancy Score

The Nancy score was validated in 2010 and, similar to the Clermont score, uses a DWI sequence, but requires a visual assessment assessing both the small and large bowel [68].

The Nancy score is calculated as follows: ulceration + parietal edema + bowel wall thickening + differentiation between the (sub) mucosa and muscularis propria + rapid contrast enhancement + DWI hyperintensity. The presence or absence of these findings is rated as one or zero (Table 4).

### 4.6. Differential Diagnoses Mimicking Crohn’s Disease on Magnetic Resonance Enterography

In the radiologic evaluation of suspected Crohn’s disease via Magnetic Resonance Enterography (MRE), a wide array of differential diagnoses must be carefully considered to avoid misdiagnosis and ensure appropriate management. Intestinal lymphoma, for instance, may present with segmental bowel wall thickening and prominent mesenteric lymphadenopathy, raising potential confusion with Crohn’s-related inflammation. Neuroendocrine tumors (NETs), although rare, can simulate Crohn’s through desmoplastic reactions causing tethering and retraction of bowel loops. Infectious etiologies such as *Mycobacterium tuberculosis*, *Yersinia enterocolitica*, and *Campylobacter* species can induce ileitis with overlapping imaging features, including mucosal ulceration and mesenteric changes [69]. Acute enteritis—whether drug-induced (e.g., NSAIDs) or ischemic—may also produce inflammatory patterns that are difficult to distinguish from early Crohn’s [70]. Of particular note, coeliac disease, though typically a malabsorptive condition, can rarely present with severe inflammation and bowel wall thickening that mimics Crohn’s disease. A remarkable case of coeliac crisis visualized on MRE has been reported in the literature underscoring the diagnostic challenge in atypical presentations. More common and specific findings of celiac disease such as inversion of the jejunoileal fold pattern and mesenteric lymphadenopathy are reviewed. More uncommon entities that are more frequently associated with refractory or untreated celiac disease, such as ulcerative jejunoileitis, cavitary mesenteric lymph node syndrome, and malignancies including small bowel adenocarcinoma and lymphoma, are described [71]. Beyond classic mimickers, other entities can present with overlapping imaging features. Ulcerative colitis with backwash ileitis may cause terminal ileal thickening, while radiation enteritis and drug-induced or ischemic enteritis can simulate the inflammatory or fibrostenotic patterns of Crohn’s [72]. Systemic conditions such as vasculitis highlights ischemic changes, multifocal wall thickening, possible hemorrhage or perforation, and eosinophilic gastroenteritis can also produce segmental wall thickening, ascites and mucosal enhancement [73]. Behçet’s disease, often involving the ileocecal region, may present with deep ulcerations, thickened segments and skip lesions, mimicking Crohn’s distribution. Additionally, small bowel neoplasms—including adenocarcinomas and gastrointestinal stromal tumors (GISTs)—can masquerade as localized stricturing disease or mass-like lesions. Post-transplant patients may exhibit graft-versus-host disease (GvHD), with diffuse mucosal damage resembling diffuse enteritis [74].

## 5. Limitations and Challenges of MRE

Although Magnetic Resonance Enterography has been established as a reference imaging method for the evaluation of inflammatory bowel diseases and presents multiple advantages, there are a number of technical and logistical limitations that must be taken into account [75]. Another important aspect is the variability of protocols among institutions, which may influence the interpretation of the obtained results [28]. The standardization of guidelines and the integration of artificial intelligence could improve the applicability of such methods in the future.

Magnetic Resonance Enterography (MRE), while invaluable in assessing inflammatory bowel disease, presents several challenges in routine clinical practice. First, accessibility and cost remain significant barriers. MRE requires high-performance equipment—ideally 1.5 T or 3 T MRI scanners—specialized software, and experienced personnel, all of which may be limited in certain healthcare centers. These constraints can restrict patient access, particularly in resource-limited settings. Furthermore, the higher costs compared to other imaging modalities like ultrasound or CT, as well as longer examination times (30–45 min), contribute to longer waiting lists in busy hospitals. To mitigate these issues, developing rapid scanning protocols and implementing clear selection criteria for patient referrals are potential solutions.

Another critical issue is the lack of standardized protocols and variability in image interpretation. Differences between centers in oral contrast dosing, scanning techniques, and interpretation criteria—especially where standardized protocols are absent—can lead to inconsistencies. Without proper correlation with clinical findings and biomarkers, there is also a risk of over- or underdiagnosis. Additionally, established scoring systems like the MaRIA, Clermont, Nancy, and London scores are not consistently applied across institutions. Developing international guidelines for MRE in IBD, akin to those for colonoscopy and CT enterography, could help address these discrepancies.

Patient preparation and compliance pose further limitations. The examination requires patients to ingest a large volume of oral contrast, remain still for extended periods, and manage potential dyspeptic symptoms, which can be particularly challenging for children, the elderly, or those with severe abdominal pain. Technical limitations, such as artifacts from respiratory motion and peristalsis, or interference from intestinal gas, can degrade image quality despite the use of antiperistaltic agents. These can cause blurring, ghosting artifacts, and loss of detail which affect the ability to detect inflammation, ulcers, or stenoses accurately. Motion correction algorithms in MRE are represented by software techniques used to reduce the effects of patient or organ motion, such as compressed sensing in modern scanners which reduces breath-hold duration; alternatively, the addition of cine sequences should be considered for assessing bowel motility (Table 5).

Moreover, MRE’s sensitivity is lower than endoscopy for detecting small, superficial lesions, early-stage disease, or subtle inflammation.

Finally, contraindications also limit MRE’s use. Patients with severe claustrophobia or non-MRI-compatible metallic implants cannot undergo the procedure. Additionally, MRE is unsuitable for acute emergencies like perforations or bowel obstructions, where CT is preferred due to faster acquisition. Proposed solutions include optimizing breathing protocols and considering alternative technologies such as wide-bore MRI scanners for claustrophobic patients.

## 6. Conclusions and Future Perspectives

MRE is an advanced imaging technique with extensive applicability in DG and the monitoring of inflammatory bowel diseases. Unlike conventional imaging methods, MRE allows for the detailed characterization of inflammation and extra-luminal complications. Therefore, it has been recommended in recent guidelines for the evaluation of patients with inflammatory bowel diseases [75]. 

### 6.1. Main Benefits of MRE

The main benefits of MRE are as follows:
➢It allows for detailed evaluation of the small intestine, and is superior to colonoscopy for endoscopically inaccessible segments;➢It distinguishes between active inflammation and fibrosis, which is an essential aspect for guiding treatment;➢It detects complications of Crohn’s disease, providing critical information for therapeutic decisions;➢It is safe for repeated use, which is ideal for the long-term monitoring of patients with IBD.

### 6.2. Research and Innovation Directions

The future of MRE is moving toward optimizing the technique and integrating advanced technologies, such as the following:
➢AI—Deep learning algorithms could automate image interpretation, reducing inter-radiologist variability and improving the accuracy of diagnosis.➢New imaging techniques may provide improved MR sequences for the early detection of inflammation and more precise differentiation between edema and fibrosis.➢The development of shorter scanning protocols will allow for faster and more accessible examinations while maintaining image quality.➢The effective utilization of serum and fecal biomarkers, such as the integration of MRE with laboratory tests (e.g., fecal calprotectin), will allow for more efficient patient monitoring.

### 6.3. AI in Magnetic Resonance Enterography (MRE): Enhancing Diagnosis and Monitoring of Inflammatory Bowel Disease

Artificial intelligence (AI) is emerging as a powerful tool in Magnetic Resonance Enterography (MRE), particularly for the diagnosis and monitoring of inflammatory bowel disease (IBD), such as Crohn’s disease. AI algorithms, especially deep learning models, are increasingly used to assist radiologists by automating image interpretation, reducing variability, and improving diagnostic accuracy, and are revolutionizing the way clinicians evaluate disease activity, predict outcomes, and personalize treatment plans [76]. Studies have shown promising results in using convolutional neural networks (CNNs) and other ML models to interpret MRE images with performance comparable to expert radiologists. However, challenges remain in data standardization, interpretability, and integration into clinical workflows [12].

#### 6.3.1. Key Applications of AI in MRE for IBD

Automated Image Segmentation and QuantificationAI algorithms, particularly deep learning models, have demonstrated proficiency in automating the segmentation of bowel structures, such as the lumen and bowel wall. Studies have reported high agreement rates between AI-based segmentation and manual methods, with some achieving 75% agreement for the lumen and 81% for the bowel wall. This automation enhances consistency and reduces inter-observer variability [77,78].Quantifying of Disease Activity and ComplicationsAI models have been developed to evaluate various disease features, including bowel wall thickness, enhancement patterns, edema, classifying disease severity, and the presence of complications like strictures or fistulas. For instance, a machine learning-based radiomic model demonstrated high accuracy in predicting the presence of intestinal fibrosis, outperforming radiologists in some cases [78].Prediction of Treatment Response and Long-Term OutcomesAI has shown promise in predicting patient responses to treatment and long-term outcomes. By analyzing imaging biomarkers and clinical data, AI models can assist in stratifying patients based on their likelihood of achieving remission or experiencing disease progression [77].Integration with Other Diagnostic ModalitiesAI’s utility extends beyond MRE. For example, in capsule endoscopy, AI models have achieved high diagnostic accuracy in identifying CD lesions, with an area under the curve (AUC) of 99%. This integration facilitates a comprehensive approach to IBD diagnosis and monitoring [12,76].

#### 6.3.2. Challenges and Future Directions

Despite the promising applications, several challenges persist:
Data Quality and Standardization: Variability in imaging protocols and patient populations can affect AI model performance.Interpretability: Understanding the decision-making process of AI models is crucial for clinical trust and adoption.Integration into Clinical Workflows: Seamless incorporation of AI tools into existing clinical practices requires careful consideration of workflow and regulatory standards.

Future research is focused on addressing these challenges, with an emphasis on developing standardized AI models that can be generalized across diverse clinical settings.

## 7. Conclusions

As technology advances and protocols become standardized, MRE has the potential to become the gold standard for small bowel imaging in inflammatory bowel disease. This non-invasive method, which is well-tolerated by patients and does not require exposure to ionizing radiation, has an overall accuracy of approximately 90%. Its integration into international clinical guidelines will improve patient management and contribute to better personalization of treatment. At the same time, multidisciplinary collaboration between gastroenterologists and imaging specialists (radiologists) is essential for further optimization of the diagnostic and treatment process of patients with inflammatory bowel disease, significantly contributing to achievement of the proposed objectives.

## Figures and Tables

**Figure 1 diagnostics-15-01540-f001:**
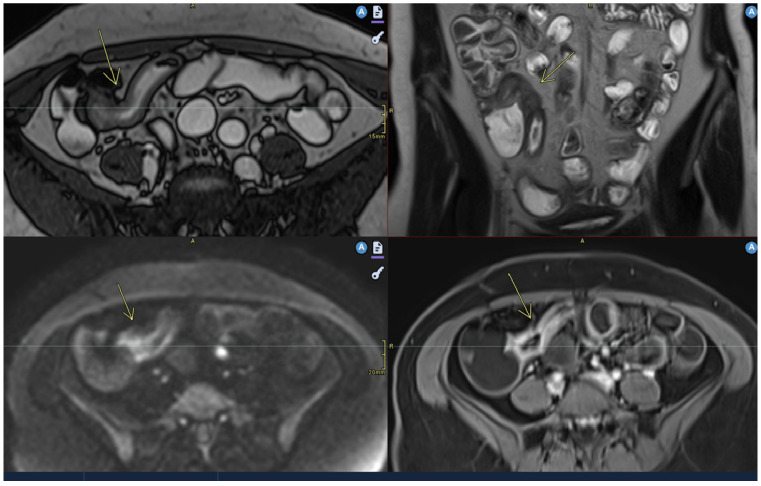
Moderate parietal thickening (7 mm). Imaging sequences include Axial T2, Coronal T2, Axial Diffusion, and Axial T1 Fat Saturation post-contrast. Findings: The last ileal loop shows moderate parietal thickening, more pronounced in the distal segment near the ileocecal valve, with a maximum thickness of 7 mm over a total length of 22 mm. This thickening is uniform, associated with diffusion restriction and early contrast enhancement, suggestive of terminal ileitis.

**Figure 2 diagnostics-15-01540-f002:**
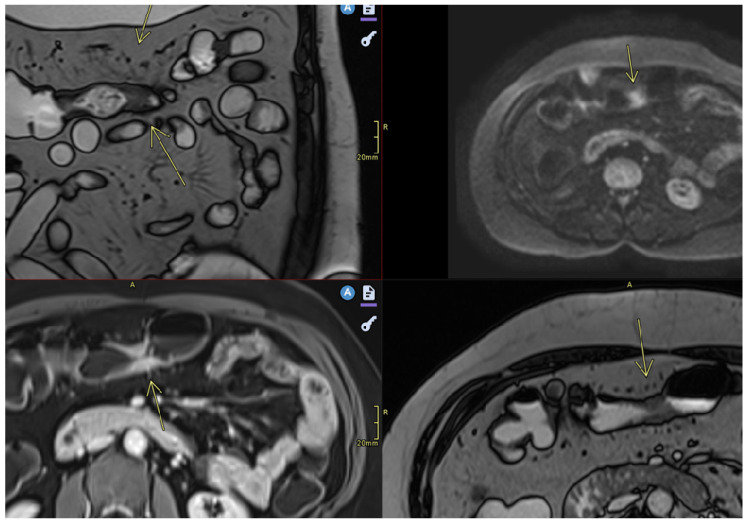
Restricted diffusion. Moderate parietal thickening. Imaging sequences include Coronal T2TRUFI (True Fast Imaging with steady-state free procession), Axial Diffusion, Axial T1 Fat Saturation post-contrast, and Axial T2TRUFI. Findings: There is circumferential parietal thickening at the level of the transverse colon, up to 8 mm in thickness over a length of 21 mm, associated with diffusion restriction and early contrast enhancement, suggestive of an acute inflammatory substrate (moderate degree of activity). Additionally, fat stranding is noted in the adjacent fat and peri-digestive vascular hyperemia is observed. No ulcerations or fistulas are present.

**Figure 3 diagnostics-15-01540-f003:**
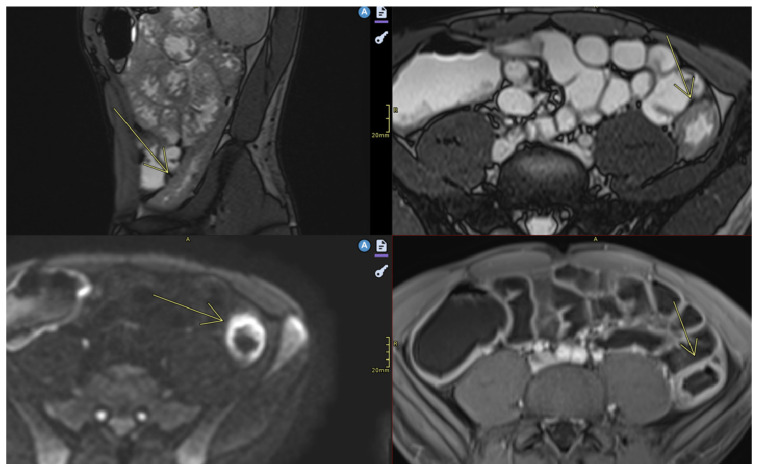
Restricted diffusion. Moderate parietal thickening (8 mm). Imaging sequences performed: Sagittal T2TRUFI, Axial T2TRUFI, Axial Diffusion, and Axial T1 Fat Saturation post-contrast. Findings: Moderate parietal thickening is noted at the level of the descending colon, associated with diffusion restriction and early contrast enhancement, suggestive of an acute inflammatory substrate. The patient presents with suspected Crohn’s disease. No ulcerations or fistulas are observed, and no marked distention of the upper rectum or sigmoid colon is present.

**Figure 4 diagnostics-15-01540-f004:**
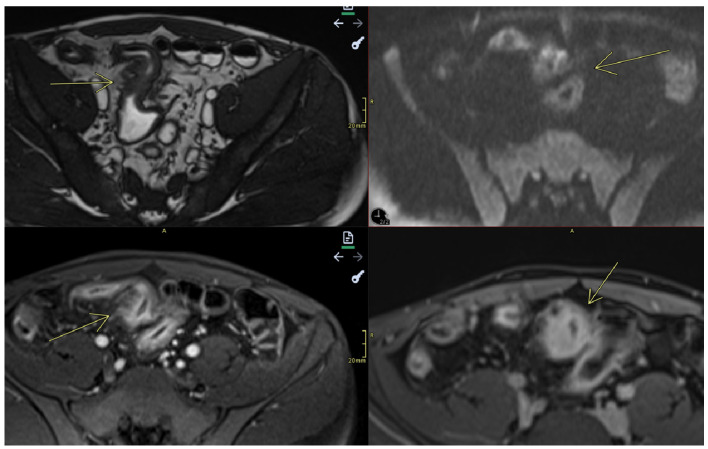
Ulcerations. Imaging sequences performed: Axial T2TRUFI, Axial Diffusion, and Axial T1 Fat Saturation post-contrast (arterial and venous phases). Findings: Multifocal ileal parietal thickening is noted, with fat stranding in the adjacent fat and peri-digestive vascular hyperemia. Minimal parietal ulcerations are present, with no fistulas observed.

**Figure 5 diagnostics-15-01540-f005:**
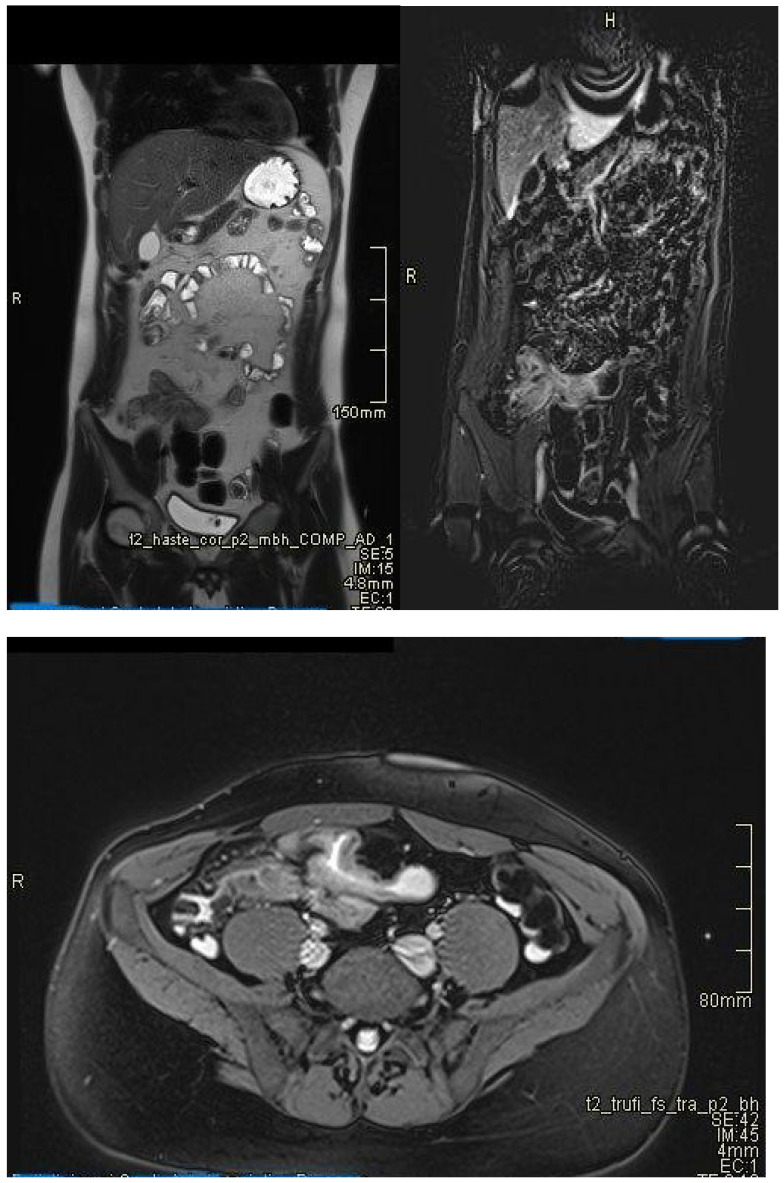
Severe wall thickening. Fibrofatty proliferation. Lymphadenopathy. Imaging sequences performed: Axial T2, T2 HASTE coronal, and T2 FS/STIR. Findings: In the right mesogastric region, several distal ileal loops are identified, displaying a “cloverleaf” configuration, with mildly circumferential wall thickening measuring up to 4 mm, suggestive of a volvulus-like appearance. The ileal segment located medially to this cloverleaf-shaped cluster demonstrates more significant parietal thickening, up to approximately 10 mm, with associated luminal narrowing. Edematous and inflammatory changes are noted in the surrounding fat. Right pericolic lymphadenopathy is present, measuring approximately 10 mm.

**Figure 6 diagnostics-15-01540-f006:**
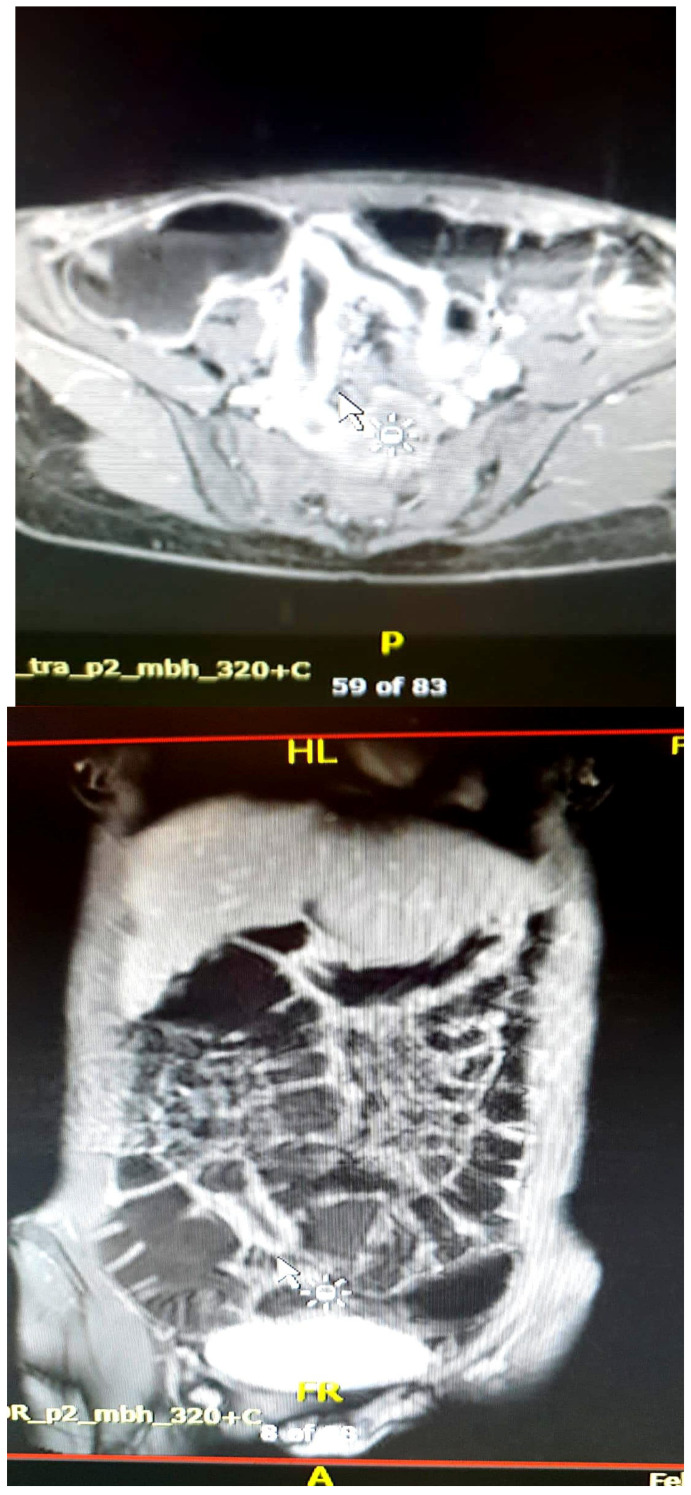
Comb sign. Imaging sequences performed: Axial T1 post-contrast Fat-sat, and Coronal T1 post-contrastfat-sat. Findings: The last segment of the ileal loop shows circumferential mucosal thickening (maximum thickness approximately 7 mm). These ileal parietal changes involve a long ileal segment located paracecal and in the right iliac fossa and hemipelvis. There is accentuation of the adjacent mesenteric vascular pattern. Upstream of this segment, with uniformly thickened and circumferential walls, several moderately distended ileal loops are evident, with a moderately visible hydro-aeric level.

**Figure 7 diagnostics-15-01540-f007:**
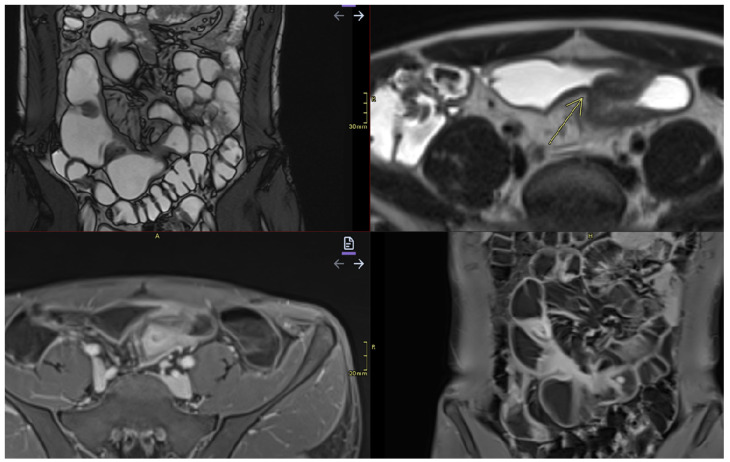
Stenosis and fistulization. Imaging sequences performed: Coronal T2TRUFI, Axial T2, and Axial and Coronal T1 Fat Saturation post-contrast. Findings: The images reveal areas of stenosis and patchy dilatations within the affected ileum, as well as areas of enteric–enteric fistulization (yellow arrow).

**Table 1 diagnostics-15-01540-t001:** MRE protocol for Crohn’s disease and ulcerative colitis.

Crohn’s Disease	Ulcerative Colitis
Intestinal preparation before examination	Colonic preparation before examination
T2-weighted HASTE/SSFSE axial and coronal	MR images in axial plane with entire colon and rectum [45]
Balanced steady state free procession gradient-echo (SSFPGR)—Coronal	T2-weighted coronal post-contrast additional images to verify if there are complications [45]
3D cinematic bSSFP—Coronal	
Delayed 3D T1-weighted post-contrast fat-saturated GRE (gradient recalled echo)—Axial	Sagittal T2-weighted MR images for anastomosis [45]
3D T1-weighted pre-/post-contrast fat-saturated GRE (gradient recalled echo)–Dynamic—Coronal	Thin-section axial fat-suppression T2-weighted images for perianal disease [45]
DWI—Axial	

**Table 3 diagnostics-15-01540-t003:** MR features for the calculations or MEGS [62,63].

Per-Segment Score				
MR features	0	1	2	3
Mural thickness	<3 mm	>3–5 mm	>5–7 mm	>7 mm
Mural T2 signal (edema)	NORMAL	Minor increase	Moderate increase	Large increase
PerimuralT2 signal	NORMAL	Increased signal but no fluid	Small (≤2 mm) fluid rim	Large (>2 mm fluid rim)
Contrast enhancement: pattern	NORMAL	N/A or homogeneous	Mucosal	Layered
Haustral loss (colon only)	0–5 cm	5–15 cm	>15 cm	
Multiplication factor for segmental score				
	X1	X2	X3	
Length of disease in that segment	<5 cm	5–15 cm	>15 cm	
Per patient score				
MR features	0		5	
Lymph nodes	Absent		Present	
Comb sign	Absent		Present	
Abscess	Absent		Present	
Fistula	Absent		Present	

**Table 4 diagnostics-15-01540-t004:** Summerize data for MRE scores.

Score Name	Validation Status	Complexity	Content/Parameters Assessed	Comments
MaRIA	Validated	High (time-consuming)	Wall thickness, relative contrast enhancement (RCE), edema, ulceration	Widely studied; requires contrast; calculation is complex and time-consuming (~12+ min).
sMaRIA (Simplified MaRIA)	Validated (2019)	Low (fast, ~4.5 min)	Wall thickness >3 mm, edema, fat stranding, ulcers	No contrast needed; simpler and faster; good sensitivity/specificity for active/severe disease.
MEGS (MR Enterography Global Score)	Validated	Moderate to High	Segmental mural thickness, mural T2 signal (edema), perimuralT2 signal, contrast enhancement pattern, haustral loss, lymph nodes, comb sign, abscess, fistula (whole bowel assessed)	Comprehensive; sums segment scores multiplied by length; includes per-patient findings; complex.
London/Extended London	Validated (2012)	Moderate	Mural thickness score, mural T2 score; extended version includes inflammation and complications	Requires contrast; mainly for research; extended version more detailed but less used clinically.
CDMI (Crohn’s Disease MRI Index)	Validated (2012)	High	Precise measurements including post-contrast gradient calculations	Rigorous and detailed with histologic correlation; requires specialized software; less clinical use.
Clermont	Validated (2013)	Moderate	Wall thickness, edema, ulceration, ADC (apparent diffusion coefficient) from diffusion-weighted imaging (DWI)	Similar toMaRIA but incorporates DWI; high specificity; requires ADC measurements.
Nancy	Validated (2010)	Moderate	Ulceration, parietal edema, bowel wall thickening, mucosa vs. muscularis differentiation, rapid contrast enhancement, DWI hyperintensity	Visual assessment of both small and large bowel; incorporates DWI; binary scoring of findings.

**Table 5 diagnostics-15-01540-t005:** Sequence Optimization.

Issue	Fix
Long scan time → more motion	Use parallel imaging (GRAPPA/SENSE) or compressed sensing
Motion sensitivity in DWI	Use readout-segmented EPI or reduced FOV DWI to reduce distortions
T2-W sequences blurred	Use single-shot fast spin echo (SSFSE/HASTE) with fat suppression and breath-hold
Ghosting in post-contrast	Use 3D T1 GRE with high acceleration, tight shim, and good pre-scan normalization

## Data Availability

Not applicable.

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
