# Peer review of "The Importance of Magnetic Resonance Enterography in Monitoring Inflammatory Bowel Disease: A Review of Clinical Significance and Current Challenges"

_diagnostics, 2025, doi:10.3390/diagnostics15121540_

Round 1
Reviewer 1 Report
Comments and Suggestions for Authors
This review addresses the role of magnetic resonance enterography (MRE) in the diagnosis and monitoring of inflammatory bowel disease (IBD), highlighting its non-invasive nature and its ability to provide detailed anatomical and functional information. The authors emphasize MRE’s advantages over other modalities, especially in evaluating Crohn’s disease, disease activity, and complications. They also discuss current limitations and future directions, including AI integration and protocol refinement.
MRE is undoubtedly a cornerstone in the modern assessment of IBD due to its safety, reproducibility, and excellent soft tissue contrast. However, I recommend the authors incorporate a brief discussion on differential diagnoses that may mimic Crohn’s disease on MRE. These include intestinal lymphoma, which often shows segmental thickening and lymphadenopathy; neuroendocrine tumors (NETs), which may present with desmoplastic reactions; and infectious etiologies such as tuberculosis (TBC), Yersinia, and Campylobacter-associated ileitis. Acute enteritis, including drug-induced and ischemic forms, can also mimic inflammatory patterns. Interestingly, a notable case report exists in the literature showing coeliac crisis visualized via MRE, and referencing that example would enrich the discussion—especially given how coeliac disease can rarely mimic Crohn’s both clinically and radiologically: https://pubmed.ncbi.nlm.nih.gov/32777280/ . These additions would not only expand the educational value of the review but also help clinicians avoid misdiagnosis in equivocal cases.
While MRE is invaluable and often preferred due to its safety profile—particularly the absence of ionizing radiation—it should not be described as the "gold standard" for IBD evaluation. That designation traditionally belongs to ballon enteroscopy or ileocolonoscopy with biopsy. MRE should rather be framed as the safest and most cost-effective cross-sectional imaging modality in this context. Adjusting the wording would enhance scientific precision without detracting from MRE’s clinical importance.
There are several areas within the manuscript that could benefit from streamlining. For example, the repeated emphasis on MRE being “non-invasive” and “free of radiation” appears in multiple consecutive paragraphs. Similarly, the phrase "current challenges" is used frequently but without adequate elaboration in each instance. A careful revision to consolidate these ideas would improve readability and make space to discuss more technical aspects, such as protocol optimization or motion artifacts.
Author Response
Thank you very much for taking the time to review this manuscript. Please, find the detailed responses below and the corresponding revisions in track changes in the re-submitted files.
Comments 1: MRE is undoubtedly a cornerstone in the modern assessment of IBD due to its safety, reproducibility, and excellent soft tissue contrast. However, I recommend the authors incorporate a brief discussion on differential diagnoses that may mimic Crohn’s disease on MRE. These include intestinal lymphoma, which often shows segmental thickening and lymphadenopathy; neuroendocrine tumors (NETs), which may present with desmoplastic reactions; and infectious etiologies such as tuberculosis (TBC), Yersinia, and Campylobacter-associated ileitis. Acute enteritis, including drug-induced and ischemic forms, can also mimic inflammatory patterns. Interestingly, a notable case report exists in the literature showing coeliac crisis visualized via MRE, and referencing that example would enrich the discussion—especially given how coeliac disease can rarely mimic Crohn’s both clinically and radiologically: https://pubmed.ncbi.nlm.nih.gov/32777280/ . These additions would not only expand the educational value of the review but also help clinicians avoid misdiagnosis in equivocal cases.
Response 1: Thank you for pointing this out. We agree with this comment. We have, accordingly, done incorporate a differential diagnoses that mimic CD in MRE – page 18, paragraph 4.7 Differential diagnoses mimicking Crohn’s Disease on Magnetic Resonance Enterography , lines between 421 and 422.
Comments 2 : While MRE is invaluable and often preferred due to its safety profile—particularly the absence of ionizing radiation—it should not be described as the "gold standard" for IBD evaluation. That designation traditionally belongs to ballon enteroscopy or ileocolonoscopy with biopsy. MRE should rather be framed as the safest and most cost-effective cross-sectional imaging modality in this context. Adjusting the wording would enhance scientific precision without detracting from MRE’s clinical importance.
Response 2: We agree with this comment. We made the following changes, according to your suggestions:
page 3, paragraph 2. Imaging methods in inflammatory bowel diseases, line115, we replaced the sentence”MRE has become the gold standard” with “MRE is essential in the evaluation of inflammatory bowel disease”.
Page 4, paragraph 2. Imaging methods in inflammatory bowel diseases, line 124 we incorporate the following sentence:” MRE is the safest and most cost-effective cross-sectional imaging method…”
Page 20, paragraph Conclusions, lines 502, 503 – we wrote:” MRE has the potential to become the gold standard…”
At page 2, paragraph Introduction , line 51 and paragraph 2. Imaging methods in inflammatory bowel diseases, line 71 , we wrote that”colonoscopy remains the gold standard for diagnosis”.
Comments 3: There are several areas within the manuscript that could benefit from streamlining. For example, the repeated emphasis on MRE being “non-invasive” and “free of radiation” appears in multiple consecutive paragraphs. Similarly, the phrase "current challenges" is used frequently but without adequate elaboration in each instance. A careful revision to consolidate these ideas would improve readability and make space to discuss more technical aspects, such as protocol optimization or motion artifacts.
Response 3: We agree with this comment. We made the following changes, according to your suggestions:
Page 2, paragraph Introduction, lines 61 and 62, we deleted the sentence” As it does not require exposure to ionizing radiation, this imaging method is preferred in young patients [2].”
Page 20, paragraph 6.2 Research and innovation directions, line 499, we changed the term “ non-invasive tests” with” laboratory tests”.
So, the terms:
“Non-invasive” remains on
- page1, paragraph Abstract, line9 ,
- page 3, paragraph 2. Imaging methods in inflammatory bowel diseases,line 87 and
- page 20, paragraph Conclusions, line 504.
“Ionizing radiation” remains on
- page 1, paragraph Abstract, line12;
- page 3, paragraph 2. Imaging methods in inflammatory bowel diseases,line 87 “radiation-free”;
- page 4, paragraph 2. Imaging methods in inflammatory bowel diseases,line 125
- Page 14, paragraph 4. Applicability of MRE in inflammatory bowel disease, line 288 – “non-irradiating”
- Page 20, paragraph Conclusions,line 505
At page 19, pargraph 6. Conclusions and future perspectives, line 477, we deleted the sentence:”…without requiring exposure to ionizing radiation”.
“Current challenges” occur on page 1, paragraph Abstract, line 13 and page 2, paragraph Introduction, line 45. We develope this idea in paragraph 6.
Regarding the protocol optimization and motion artifacts, we already wrote in paragraph 3, but we elaborated in the paragraph 5.
“ Technical limitations, such as artifacts from respiratory motion and peristalsis, or interference from intestinal gas, can degrade image quality despite the use of antiperistaltic agents. These can cause blurring, ghosting artifacts and loss of detail which affect the ability to detect inflammation, ulcers or stenoses accurately. Motion correction algorithm in MRE is represented by a software techniques used to reduce the effects of patient or organ motion such as compressed sensing in modern scanners which reduces breath-hold duration or consider adding cine sequences for assessing bowel motility.
Sequence Optimization
|
Issue |
Fix |
|
Long scan time → more motion |
Use parallel imaging (GRAPPA/SENSE) or compressed sensing |
|
Motion sensitivity in DWI |
Use readout-segmented EPI or reduced FOV DWI to reduce distortions |
|
T2-W sequences blurred |
Use single-shot fast spin echo (SSFSE/HASTE) with fat suppression and breath-hold |
|
Ghosting in post-contrast |
Use 3D T1 GRE with high acceleration, tight shim, and good pre-scan normalization” |
Reviewer 2 Report
Comments and Suggestions for Authors
Dear Editor
This is a review article regarding MRE. The followings are my comments.
1. Please remove the patient information , including ID, examination date of the illustrated MRE images, improve quality of Figure 5 and 6
2. Please re-write section 5 into paragraph rather than bullet points. The section "enteroMR" should be MRE
3. Provide a summarize table for section 4 regarding MRE scores, including whether the score was validated, complexity, content of the scores for better understanding.
4. Please provide a short review of AI in the filed of MRE.
Author Response
Thank you very much for taking the time to review this manuscript. Please, find the detailed responses below and the corresponding revisions in track changes in the re-submitted files.
Comments 1: Please remove the patient information , including ID, examination date of the illustrated MRE images, improve quality of Figure 5 and 6
Response 1 : Thank you for pointing this out. We have erased the ID of the patient . There are details of the device and the imaging sections. This changes can be found at page 11, paragraph 3.2. Relevant imaging features in Magnetic Resonance Enterography , lines 243 and 244.
At the same time, we improve the quality of Figure 5 and 6. This changes can be found at page 12 and 13 paragraph 3.2. Relevant imaging features in Magnetic Resonance Enterography , lines 256 and 257.
Comments 2: Please re-write section 5 into paragraph rather than bullet points. The section "enteroMR" should be MRE.
Response 2: Thank you for pointing this out. We re-wrote section 5 into paragraph. This change can be found at pages 18 and 19, paragraph 5. Limitations and challenges of MRE and between lines 431 to 472. Also, we corrected enteroRM with MRE on pages 18 and 19, paragraph 5. Limitations and challenges of MRE- line 422 and paragraph 6.1 Main benefits of MRE, lines 479 and 480.
Comments 3: Provide a summarize table for section 4 regarding MRE scores, including whether the score was validated, complexity, content of the scores for better understanding.
Response 3: We have , accordingly done the summerize table for section 4 to emphasize MRE scores with there validations, complexity and content at page18, paragraph 4.6 Role in Staging and Imaging Scores, after line 421.
Comments 4: Please provide a short review of AI in the filed of MRE.
Response 4: We have , accordingly done a short review of AI in the field of MRE – page 20, pargraph 6.3. AI in Magnetic Resonance Enterography (MRE): Enhancing Diagnosis and Monitoring of Inflammatory Bowel Disease, between lines 500 and 501.

Round 2
Reviewer 2 Report
Comments and Suggestions for Authors
Dear Editor,
The authors have adequately addressed all questions. I have no further concerns. I would only suggest improving the layout of the manuscript during the final proof-editing stage.